# High Oxytocin Receptor Expression Linked to Increased Cell Migration and Reduced Survival in Patients with Triple-Negative Breast Cancer

**DOI:** 10.3390/biomedicines10071595

**Published:** 2022-07-05

**Authors:** Huiping Liu, Markus Muttenthaler

**Affiliations:** 1Institute for Molecular Bioscience, The University of Queensland, Brisbane, QLD 4072, Australia; huiping.liu@uqconnect.edu.au; 2Key Laboratory of Molecular Pharmacology and Drug Evaluation, Ministry of Education, School of Pharmacy, Yantai University, Yantai 264005, China; 3Institute of Biological Chemistry, Faculty of Chemistry, University of Vienna, 1090 Vienna, Austria

**Keywords:** oxytocin receptor, epidermal growth factor (EGF), triple-negative breast cancer, metastasis, biomarker

## Abstract

Triple-negative breast cancer (TNBC) is an aggressive breast cancer subtype with limited treatment options and high mortality. The oxytocin receptor (OTR) is a class-A G protein-coupled receptor that has been linked to breast cancer, but its role in tumorigenesis and disease progression remains underexplored. OTR expression is highest in tumour-adjacent breast tissue, followed by normal and tumour tissue, indicating a potential role in the tumour microenvironment. OTR levels were higher in migrated MDA-MB-231 cells than in the control parental cells cultured in normal medium; OTR overexpression/knock-down and metastasis biomarker experiments revealed that high OTR expression enhanced metastasis capabilities. These findings align well with data from a murine breast cancer metastasis model, where metastasised tumours had higher OTR expression than the corresponding primary tumours, and high OTR expression also correlates to reduced survival in TNBC patients. OTR agonists/antagonists did not affect MDA-MB-231 cell migration, and pharmacological analysis revealed that the OT/OTR signalling was compromised. High OTR expression enhanced cell migration in an OTR ligand-independent manner, with the underlying mechanism linked to the EGF-mediated ERK1/2-RSK-rpS6 pathway. Taken together, high OTR expression seems to be involved in TNBC metastasis *via* increasing cell sensitivity to EGF. These results support a potential prognostic biomarker role of OTR and provide new mechanistic insights and opportunities for targeted treatment options for TNBC.

## 1. Introduction

Breast cancer is a biologically and molecularly heterogeneous disease comprising several subtypes with distinct biological signatures that affect clinical prognosis and treatment [1]. It has become the most commonly diagnosed cancer with an estimated 2.3 million new cases (11.7%) in 2020 [2]. Progress in clinical management strategies and earlier detection through increased awareness and widespread use of mammography has improved overall survival for female breast cancer patients, with 5-year relative survival rates of 89% [3]. Despite these positive developments, the 5-year relative survival rate for metastatic breast cancer remains at only 27%, highlighting the need for new therapeutic approaches [4].

Metastasis is a complex multistep process, including acquisition of invasive properties through genetic and epigenetic alterations, local infiltration of tumour cells into the adjacent tissue, intravasation, survival in circulation, extravasation, and metastatic colonization in distal organs [5,6,7]. Many signalling pathways have been implicated in such cancer cell migration and epithelial-to-mesenchymal transition (EMT) at the early stages of cancer cell dissemination [5,6], such as the TGFβ pathway, the poly(ADP-ribose) polymerase pathway, and the receptor tyrosine kinase (RTK) pathway [8]. Substantial efforts have been made towards the development of novel anti-metastatic targeted therapies [9], with limited success however, especially for patients with triple-negative breast cancer (TNBC).

TNBC lacks the estrogen receptor (ER), the progesterone receptor (PR), and the human epidermal growth receptor 2 (HER2) and accounts for 10–20% of all breast cancer cases. It is one of the most aggressive subtypes with a high probability of metastasis, recurrence, and development of chemotherapy resistance [10,11]. TNBC has the worst prognosis and distant metastasis-free survival among all breast cancers. TNBC patients typically do not respond to the available targeted treatments, and systemic chemotherapy remains the mainstay of clinical treatment despite causing severe side effects [11]. It is therefore critical to explore novel targets and therapeutic strategies for treating TNBC.

The oxytocin receptor (OTR) might be such a novel target. OTR is a class-A G protein-coupled receptor (GPCR) expressed in various tissues, mediating a broad range of biological functions [12]. OTR’s primary function in the female mammary gland is to mediate the effect of the endogenous peptide ligand oxytocin (OT) during lactation. Emerging evidence suggests potential links with multiple cancer types, including breast cancer [13,14,15,16]. OTR mRNA and proteins are expressed in most breast cancer cell lines and 80–90% of breast carcinomas, including TNBC tumours [17,18,19,20,21], supporting OTR as a potential target and biomarker for breast cancer. Proliferation studies support the assertion that OTR modulates breast tumour growth; however, the exact signalling pathways remain poorly understood [21]. These studies render OTR a promising, yet underexplored target for receptor-based therapy that warrants further study. This work therefore investigated the role of the OT/OTR signalling system in breast cancer with a specific focus on TNBC due to its aggressive behaviour and limited treatment options.

## 2. Materials and Methods

### 2.1. Gene Expression and Survival Analysis

The condition search tool (Perturbations) from Genevestigator (https://genevestigator.com/gv/, accessed on 7 June 2020) was used to identify conditions that significantly affect OTR expression (gene symbol: *OXTR*) [22]. Human samples (organism: *Homo sapiens*) based on the Affymetrix Human Genome U133 Plus 2.0 Array platform were selected, followed by manual selection of the studies containing normal and breast tumour tissues. The original datasets from the selected studies were downloaded from the Gene Expression Omnibus (GEO) (http://www.ncbi.nlm.nih.gov/geo/, accessed on 7 June 2020) database and analysed.

The datasets containing breast tumour and paired adjacent noncancerous tissue were retrieved by a search of the GEO database [23], using the following keywords: “breast cancer”, “tissue”, “expression profiling by array”, “adjacent”, and “paired”. The Kaplan–Meier plotter (KM-plotter, http://www.kmplot.com, accessed on 21 April 2021) for breast cancer (mRNA gene chip) was used to assess the effect of the OTR gene level on TNBC patient relapse-free survival (RFS) [24].

### 2.2. Chemicals

OT was synthesised with a purity of >98%, analysed by high-performance liquid chromatography (HPLC). EGF recombinant human protein was purchased from Life Technologies Australia Pty Ltd. Invitrogen (PHG0311, Mulgrave, VIC, Australia). Gefitinib was purchased from Sigma Aldrich (SML-1657-10MG, Bayswater, VIC, Australia). U0126 was purchased from Cell Signaling Technology (9903S). Insolution™ Rapamycin was purchased from Merck Pty Ltd. (US1553211-1MG, Kilsyth, VIC, Australia). Gefitinib, rapamycin, and U0126 were dissolved and stocked in DMSO.

### 2.3. Cells and Cell Transfection

The human breast cancer cell line MDA-MB-231 is a TNBC cell line originally established from the pleural effusion of a female patient with breast adenocarcinoma. The cell line was a gift from A/Prof Andreas Möller (QIMR Berghofer Medical Research Institute). The cell line was maintained in DMEM medium supplemented with 10% FBS at 37 °C in a humid atmosphere with 5% CO_2_ and 95% air, routinely confirmed negative for mycoplasma and bacteria contamination, and the cell line was authenticated using STR profiling. The human OTR expression plasmid (pCMV6-OTR, RC211797) and the empty vector (pCMV6-vector, PS100001) were purchased from Origene (Rockville, MD, USA). The human G-protein alpha 15 subunit (wild-type, GNA15), cloned into pcDNA3.1 (pcDNA3.1-GNA15), was obtained from the cDNA Resource Center (www.cdna.org, Bloomsberg, PA, USA). Plasmid transient transfections were carried out with FuGENE HD transfection reagent (Promega, Auburn, VIC, Australia). Selective siRNAs to knock down human OTR (M-005688-02-0005) and the control (non-targeted) siRNA (D-001206-13-05) were purchased from Dharmacon as SMARTpools. OTR knock-down transfection was carried out at a 25 nM final concentration of siRNA with DharmaFECT 1 transfection reagent (Dharmacon, Lafayette, CO, USA). The OTR overexpression and knock-down efficiency tested by qPCR are shown in Appendix A.

### 2.4. Transwell Cell Migration Assays

To separate and amplify migrated cells, the cell migration assay was carried out using 24-well transwell inserts (8 μm pore size). The cells were starved for 24 h, harvested, and suspended in serum-free DMEM with 0.1% BSA. The cells were loaded into the upper chambers (2 × 10^5^ cells in 0.1 mL/insert) and incubated for ~2 h at 37 °C. Then, control and chemo-attractants (10% FBS) were loaded into the lower chamber (650 µL/well) and incubated for 24 h. The cells that migrated to the bottom of the membrane and the chamber were collected (T1 cells). After amplification under normal cell culture conditions, the amplified T1 cells were admitted to the next cycle of migration and amplification. The 24-well transwell migration assay was also applied for assessing 10% FBS- and 20 ng/mL epidermal growth factor (EGF)-guided chemotaxis. After 24 h migration, the cells inside the inserts were gently removed using wet cotton swabs; the filter membrane was washed with PBS, fixed with 70% ethanol, and stained with 0.2% crystal violet. The number of migrating cells was counted manually under a light microscope.

To evaluate the effects of OTR agonists/antagonists on the migration ability of the MDA-MB-231 cells, HTS Transwell^®^-96 permeable supports (8 μm pore size) were applied for higher throughput using 10% FBS as a chemoattractant. The cells were loaded into the upper chambers (5 × 10^4^ cells in 50 µL/insert) and incubated for ~2 h at 37 °C. Then, OTR agonists/antagonists were added to the inserts to a final concentration of 50 nM; the control and chemoattractant (10% FBS) were loaded into the receiver plate (150 µL/well) and incubated for 24 h. After 24 h migration, the migrated cells were stained with crystal violet as described above. The bound crystal violet was eluted by adding 150 μL of 33% (*v*/*v*, diluted with ddH_2_O) acetic acid into the receiver plate and shaking for 10 min. The eluent (100 μL) from the receiver plate was transferred to a transparent 96-well plate, and the absorbance at 570 nm (OD570) was measured on an INFINITE M1000 PRO plate reader (Tecan Austria GmbH, Grödig, Austria). A standard curve of cell number vs. OD570 was plotted for quantification (Appendix A).

### 2.5. qPCR

TRIzol reagent was used for total RNA extraction. cDNA was synthesised from total RNA (500 ng/sample) using Invitrogen SuperScript™ III Reverse Transcriptase (Thermo Fisher Scientific, Waltham, MA, USA). Applied Biosystems TaqMan Gene Expression assays were used for quantitative real-time PCR analysis of gene expression. Relative mRNA levels of OTR (Hs00168573_m1) and E-cadherin (Hs01023895_m1) were compared with the housekeeping gene GAPDH (Hs02786624_g1). The ^ΔΔ^CT Fast Taqman method was performed to analyse gene expression.

### 2.6. Scratch Wound Healing Assay

Cells (2 × 10^5^ cells/well) were grown in 24-well plates overnight and then serum-starved for 24 h. Then, the cells were scratched using 200 μL pipette tips, immediately rinsed twice with PBS, and subsequently cultured in serum-free medium in the presence of the treatment or vehicles. Photos were taken using a Nikon Ti-U inverted fluorescence microscope at 0 h and 24 h after the scratches were made (20× objectives). The wound width was measured using Fiji/ImageJ.

### 2.7. Gelatin Zymography Assay

The cells were grown in 6-well plates (2 × 10^5^ cells/well/2 mL). The medium was removed after adhesion, and the cells were washed twice with serum-free media and then incubated in serum-free medium for 24 h in the presence of the ligand treatment or vehicle. The medium in the plate after 24 h incubation was the conditioned medium for the gelatin zymography assay, an indirect measure of MMP-9 expression. The samples (30 μg/sample) were fractionated on SDS-PAGE gel containing 1 mg/mL gelatin in the resolving gel. The gels were washed 2 × 30 min with washing buffer (2.5% Triton X-100, 50 mM Tris-HCl, pH 7.5, 5 mM CaCl_2_, 1 μM ZnCl_2_), rinsed for 5–10 min in incubation buffer (1% Triton X-100, 50 mM Tris-HCl, pH 7.5, 5 mM CaCl_2_, 1 μM ZnCl_2_) at 37 °C, and then incubated in fresh incubation buffer for 24 h at 37 °C with agitation. The gels were then stained with 0.5% Coomassie blue for 30 min and destained until bands could clearly be seen. Non-stained regions of the gel corresponding to gelatinase activity were quantified by densitometry using the BIO-RAD ChemiDoc^TM^ Imaging System (Bio-Rad Laboratories, Hercules, CA, USA).

### 2.8. Western Blots

The cells were seeded in 48-well plates (~10 × 10^4^ cells/well) and serum-starved overnight before treatment. The cells were washed with ice-cold PBS after corresponding treatments and then lysed with Laemmli buffer containing 50 mM DTT (Sigma-Aldrich, Bayswater, VIC, Australia). Total protein samples (~20 μg/sample) were fractionated by SDS-PAGE gel electrophoresis and transferred onto Polyvinylidene Fluoride (PVDF) membrane (Merk Millipore Ltd., County Cork, Ireland). The membrane was probed with different primary antibodies, including mouse AKT/MAPK Signaling Pathway Antibody Cocktail (ab151279; Abcam), Phospho-S6 Ribosomal Protein (Ser^235/236^) Antibody (2211; Cell Signaling Technology), and mouse anti-GAPDH antibody (sc-47724; Santa Cruz). The membranes were then incubated with the appropriate IRDye680 or IRD800 labelled secondary antibody, imaged, and analysed using the ODYSSEY Infrared Imaging System (LI-COR, Lincoln, NE, USA).

### 2.9. Fluorescence Imaging Plate Reader Functional Calcium Assay

Activation of G_q_-signalling after OTR ligand stimulation in MDA-MB-231 cells was measured by detection of intracellular calcium release using a fluorescent imaging plate reader (FLIPR) Calcium 4 Evaluation kit (Molecular Devices, Sunnyvale, CA, USA). An SH-SY5Y cell line with native OTR expression and good G_q_/calcium response was used as the positive signal control for the FLIPR assay. Briefly, the cells were plated in 384-well plates with a black wall and clear bottom and incubated at 37 °C overnight. The next day, the supernatant was removed, and the cells were incubated in a calcium dye loading buffer for 30 min at 37 °C. Then, the plate was transferred to a FLIPR Tetra instrument (Molecular Device, USA) for peptide injection and fluorescence measurements. The OTR ligands were added at various concentrations once reading commenced, and fluorescence was measured in real time from the bottom of the plate.

### 2.10. Functional cAMP Assay

The PerkinElmer’s LANCE^®^ Ultra cAMP kit (PerkinElmer, Melbourne, VIC, Australia) was applied to measure cAMP produced via activation of G_s_-signalling in MDA-MB-231 cells upon OTR ligand stimulation. The assay was conducted in 384-well plates with a total volume of 20 µL. For the OTR-overexpressing group, cAMP accumulation measurements were performed 48 h after transient plasmid transfection. Briefly, the peptides and cells were prepared in freshly made stimulation buffer from the kit. After the peptide dilutions were prepared and added into the 384-well plate, the cells were added and incubated at 37 °C for 30 min. Following the incubation, the Eu-cAMP tracer and the U*Light*-cAMP mAb prepared in the cAMP detection buffer provided within the kit were added into the plate, followed by incubation at 25 °C for 1 h. The TR-FRET signal was then detected at 665 nm on an INFINITE M1000 PRO plate reader (Tecan Austria GmbH, Grödig, Austria).

### 2.11. Statistical Analysis

All data were expressed as the mean ± standard error of the mean (SEM) of at least three independent assays unless otherwise specified. The data were analysed using Prism 9 (GraphPad Software Version 9.0.0, San Diego, CA, USA). ANOVA analysis followed by Tukey’s multiple comparisons test was used when comparing more than two groups. The significance level was considered as below 0.05 in all experiments.

## 3. Results

### 3.1. Oxytocin Receptor Expression Is Highest in Tumour-Adjacent Breast Tissues Followed by Normal and Tumour Breast Tissue

Only limited information exists regarding OTR expression in healthy breast compared to breast cancer tissue, which is, however, central for understanding OTR’s functional role in breast cancer and for supporting potential OTR-specific therapeutic strategies. Ten datasets containing both breast tumour tissue and normal breast tissues were identified from Genevestigator (Figure 1A,B, Table 1). The OTR gene expression was significantly higher in normal breast tissues than in breast tumour tissues in 7 out of 10 datasets (Figure 1B), including GSE21422 (*p* = 0.0434, normal vs. invasive ductal carcinomas), GSE7904 (*p* < 0.0001, normal vs. tumour), GSE10810 (*p* = 0.0007, normal vs. tumour), GSE31448 (*p* = 0.0003, normal vs. tumour), GSE10780 (*p* < 0.0001, normal vs. tumour), GSE20711 (*p* = 0.0010, normal vs. tumour), and GSE3744 (*p* < 0.0001, normal vs. tumour).

Additionally, OTR gene expression in paired adjacent and tumour breast tissues was analysed from three datasets retrieved from the GEO database (Table 1), including GSE109169, containing 25 paired adjacent and tumour samples (no ER/PR/HER2 expression data available), GSE139038, containing 18 paired samples (including two TNBC tumour tissue with paired adjacent tissue), and GSE76250, containing 33 paired adjacent breast tissues and the corresponding TNBC tissues and 132 non-paired TNBC tumour tissues. This analysis confirmed that OTR mRNA levels are higher in adjacent breast tissue compared to the paired/corresponding breast tumour tissue (Figure 1C).

### 3.2. Oxytocin Receptor Expression Is Higher in Migrated/Metastasised Breast Cancer Cells Than in Primary Cells

Chemotactic cell migration is a fundamental mechanism underlying cancer cell metastasis. To determine whether OTR expression correlates with chemotaxis of breast cancer cells, the OTR levels in migrated cells were analysed (Figure 2A,B). The MDA-MB-231 cell line was chosen due to its well-studied characteristics, relevance in metastasis, good translation for in vivo models, and its triple-negative subtype [25,26]. After two and four screening and amplification cycles, the 2nd (T2) and 4th (T4) generation of cells had ~2.1- and 2.5-fold higher OTR mRNA expression compared to the control parental cells cultured in a normal medium with 10% FBS, respectively (*p* < 0.0001, Figure 2B).

These results aligned well with the dataset GSE137842, derived from an in vivo study mimicking human bone metastasis in mice using MDA-MB-231 cells [27]. The OTR expression was higher in the tumour cells that metastasised in human bone implants than in their corresponding primary xenograft tumour cells analysed by GEO2R (Figure 2C, log2-fold change = 2.0586, adjusted *p* = 0.0146) [27]. These data point to a correlation between high OTR levels and metastasis in triple-negative MDA-MB-231 cells both in vitro and in vivo. Such a correlation was further confirmed by the migration capability results from MDA-MB-231 cells with OTR overexpression or knock-down. When OTR was overexpressed (pCMV6-OTR group), more cells migrated to the bottom well than the vector group (Figure 2D, *p* < 0.01). When OTR was knocked down (siOTR group), fewer cells migrated than the non-targeting control RNA group (Figure 2E, *p* < 0.05).

Patient survival analysis showed that RFS was lower for TNBC patients with high OTR expression in the tumour tissues than those with low OTR expression (Figure 2F, *p* = 0.0047), aligning well with the data analysis and the in vitro results described above and supporting the hypothesis that high OTR expression may play a role in facilitating metastasis, which is directly linked to reduced patient survival.

### 3.3. Oxytocin Receptor Agonists/Antagonists Do Not Affect Cell Migration, and the Oxytocin Receptor Does Not Function via G_q-_- and G_s_-Pathways in MDA-MB-231 Cells

As OTR expression correlated with cell migration, it was of interest to investigate if OTR agonists/antagonists would affect cell migration. Four peptides were selected based on their pharmacological properties (Figure 3A) [28,29]: endogenous OT, the selective OTR agonist [Thr^4^,Gly^7^]OT (TGOT), the selective OTR antagonist desGly-NH_2_,d(CH_2_)_5_[D-Tyr^2^,Thr^4^]OVT, and the approved drug atosiban (AT), which is a biased OTR ligand that blocks the G_q_ pathway and activates the G_i3_ pathway [30]. None of the OTR agonists/antagonists significantly affected cell migration in control or OTR-overexpressing MDA-MB-231 cells (Figure 3B).

As no significant differences in cell migration were observed between the OTR agonists and antagonists, OTR signalling was investigated. OTR activation typically results in G_q_-mediated calcium accumulation, including in breast cancer cells [19,31,32]. One study also reported G_s_-mediated cyclic adenosine monophosphate (cAMP) accumulation in MDA-MB-231 cells upon 100 nM OT treatment [33]. MDA-MB-231 cells also express the vasopressin (VP) V_2_R that is activated by OT at higher concentrations and that signals via the G_s_-cAMP pathway [34,35]. Therefore, both calcium and cAMP accumulation were examined to determine whether OTR was functional in the MDA-MB-231 cells or if there might be an involvement of the VP receptors.

Interestingly, no intracellular calcium accumulation was detected upon OT stimulation regardless of whether or not there was OTR overexpression (Figure 3C). EGF also did not induce a significant signal in the FLIPR assay at the concentration of 20 ng/mL, as used in the migration assay. A vector encoding for GNA15, a member of the G_q_ subunit family that mediates intracellular signalling for calcium release [36], was co-transfected with the pCMV6-OTR plasmid to achieve overexpression of both GNA15 and OTR in MDA-MB-231 cells. This experiment would reveal if the lack of signal was due to a general lack or downregulation of G_q_ [37]. However, again, no significant intracellular calcium accumulation upon OT treatment in the GNA15- and OTR-overexpressing cells was observed (Figure 3C), supporting that OTR was not functional and that this was independent of G_q_. These findings also excluded the involvement of V_1a_R and V_1b_R, which are activated by OT at the tested concentrations [35,38] and that also signal via the G_q_/calcium pathway. OT or VP (1 pM–10 μM) also did not induce any intracellular cAMP accumulation (Figure 3D), thereby excluding a functional OTR- or V_2_R-mediated G_s_ pathway.

These results confirmed that OTR was not functional in the classical sense of G_q_/G_s_-protein dependent signalling in MDA-MB-231 cells. This is interesting as a compromised OTR function could be a cancer-promoting mechanism undermining the protective role of OTR against breast cancer development, as discussed in another study [39].

### 3.4. MDA-MB-231 Cells Overexpressing the Oxytocin Receptor Are More Sensitive to EGF

The EMT process is a critical driver of TNBC metastasis, and EGF is one of the most abundant growth factors in the tumour microenvironment inducing the EMT during local invasion. FBS contains multiple growth factors including EGF; for example, the concentration of betacellulin (a member of the EGF family) is ~3.68 ng/mL in FBS [40]. OTR overexpression-induced metastasis might thus be associated with EGF in the tumour microenvironment, which was investigated by replacing 10% FBS with EGF (20 ng/mL) as the chemoattractant in the transwell migration assay. Reducing the complexity of the chemoattractant makes it EGF-specific, supporting further pathway analysis.

OTR up- or down-regulation still led to increased or decreased cell migrations with EGF as the sole chemoattractant (*p* < 0.05, Figure 4A,B), indicating that EGF plays a key role and OTR overexpression increases cell sensitivity to EGF-mediated migration. This EGF-mediated migration ability of cells with or without OTR overexpression was further confirmed in scratch wound healing assays, where the cell monolayer gap closed faster upon EGF stimulation in the pCMV6-OTR group than in the vector group (Figure 4C).

E-cadherin is a predictive marker for breast cancer metastasis, as loss of E-cadherin facilitates metastasis formation by disrupting intercellular contacts, an early step of metastatic dissemination. Matrix metalloproteinases (MMPs) are a family of zinc-dependent endoproteinases that catalyse hydrolysis of extracellular matrix (ECM) components, thereby facilitating tumour cell invasion and metastasis. EGF significantly decreased E-cadherin in OTR-overexpressing MDA-MB-231 cells (Figure 4D, *p* < 0.05), and E-cadherin expression was also significantly lower in the EGF-treated pCMV6-OTR group than the EGF-treated vector group (Figure 4D, *p* < 0.05). In addition, the MMP-9 expression was promoted (Figure 4E), which was confirmed by gelatin zymography assay. However, the differences of MMP-9 expression between the EGF-treated vector group and the EGF-treated pCMV6-OTR group did not reach statistical significance. Addition of OT (1 μM) to EGF (20 ng/mL) did not affect cell migration in the scratch wound healing assay (Figure 4C), and the EGF-stimulated upregulation of E-cadherin and MMP-9 was also not significantly affected (Figure 4D,E). These results support an OTR-enhanced, EGF-stimulated migration that is independent of OTR ligand activation.

### 3.5. Oxytocin Receptor Enhances EGF-Stimulated RSK Activation, with the mTOR Pathway Contributing to the Downstream rpS6 Activation

The ERK (extracellular signal regulated kinase) and mTOR (mammalian target of rapamycin) pathways are recognised key coordinators of the EMT process and cell migration during cancer metastasis, and both pathways can mediate the intracellular signals of OTR [12,41] and EGFR [42,43] independently (Figure 5A). RSK (ribosomal s6 kinase) is a principal effector of ERK that promotes the motility and invasive capacity of carcinoma cells [44], and RSK can also activate mTOR [45]. RpS6 (ribosomal protein S6) is another protein of interest; it is a downstream effector of both the ERK and the mTOR pathways [46] (Figure 5A), and it is essential for the protein synthesis required during cancer progression. Phosphorylated ERK1/2 (p-ERK1/2), RSK (p-RSK), and rpS6 (p-rpS6) were therefore measured to investigate the intracellular signalling mechanism underlying the EGF-stimulated migration of OTR-overexpressing MDA-MB-231 cells. The phosphorylation of ERK1/2, RSK, and rpS6 all peaked between 30 and 60 min of treatment in both the OTR-overexpressing and the control cells (Figure 5B). This is most likely due to EGF binding to EGFR on the cell surface, which usually peaks after 30–40 min and declines once EGF is internalised and degraded [47]. EGF induced similar ERK1/2 and rpS6 phosphorylation in the OTR-overexpressing and control cells but significantly more RSK phosphorylation in the OTR-overexpressing cells at 30 and 60 min (Figure 5B, *p* = 0.0111 and *p* = 0.0065).

To verify the involvement of EGFR in EGF-induced ERK1/2-RSK-rpS6 activation, the cells were pre-treated with the EGFR inhibitor gefitinib (20 μM) one hour prior to stimulation with EGF, and gefitinib remained in the solution with subsequent EGF treatment for up to four hours. In the control cells, gefitinib completely blocked ERK1/2, RSK, and rpS6 phosphorylation (Figure 5B). By contrast, in OTR-overexpressing cells, gefitinib only partially inhibited the EGF-stimulated ERK1/2 and RSK phosphorylation, with a significant phosphorylation peak difference at 60 min (Figure 5B, *p* < 0.01 and *p* < 0.05 pCMV6-OTR vs. vector group at 60 min for ERK1/2 and RSK, respectively). This indicates that EGF can induce ERK1/2 and RSK phosphorylation in an EGFR-independent manner. Moreover, gefitinib failed to inhibit the OTR-overexpressing cell migration (Figure 5C, *p* > 0.05 gefitinib vs. vehicle), confirming that EGF was able to induce EGFR-independent but OTR-dependent migration when OTR was overexpressed.

More downstream-pathway-specific inhibitors were used to dissect the pathways further. U0126 is a selective small molecule inhibitor for MEK1/2, the immediate upstream kinase of ERK1/2 [48]. In contrast to gefitinib, U0126 (10 μM) completely blocked the EGF-induced phosphorylation of ERK1/2 and the downstream RSK and rpS6 in the OTR-overexpressing cells (Figure 5D). No obvious differences of the inhibition efficacy of U0126 were observed between the control cells and the OTR-overexpressing cells (Figure 5D). The U0126-treated cells migrated less than untreated cells in both the control and the OTR-overexpressing cells (Figure 5C, *p* < 0.001 and *p* < 0.01 U0126 vs. vehicle in vector and pCMV6-OTR groups, respectively). Furthermore, U0126 successfully inhibited the cell migration of the OTR-overexpressing cells (*p* < 0.01). As U0126 specifically inhibits the MEK–ERK1/2 pathway, in contrast to gefitinib, which blocks EGFR further upstream, it seems that the MEK-ERK1/2 pathway plays a dominant role in EGF signalling but is not entirely dependent on EGFR when OTR is overexpressed.

Both the mTOR and the ERK1/2-RSK pathways are independently linked to rpS6 activation [46]. To assess whether the mTOR pathway was involved, the OTR-overexpressing MDA-MB-231 cells were pre-treated with the selective mTOR inhibitor rapamycin (5 nM) for one hour before EGF was added. Rapamycin did not affect ERK1/2 activation compared to cells only treated with EGF, but blocked rpS6 phosphorylation (Figure 5D). Rapamycin also reduced EGF-stimulated RSK activation, but the effects of rapamycin on RSK activation in the OTR-overexpressing MDA-MB-231 cells were not further investigated. Rapamycin also decreased the cell migration of OTR-overexpressing cells but with lower efficacy than U0126 (Figure 5C). These results suggest that in OTR-overexpressing cells, the mTOR pathway contributes to rpS6 phosphorylation upon EGF stimulation, independently of ERK1/2 activation, indicating that mTOR and ERK1/2 pathways that are differentially able to activate the downstream rpS6 have different metastatic potential.

## 4. Discussion

TNBC remains the most difficult breast cancer subtype to treat, with poorer clinical outcomes than the other subtypes [10,11]. TNBC lacks targeted therapy approaches, leaving chemotherapy as the mainstay of treatment options [11]. OTR has been implicated in breast cancer initiation and progression [13,14,15,16,17,20], but its mechanistic role in breast cancer development and progression remains underexplored. This study investigated OTR’s expression profile in breast cancer patients and revealed that OTR can facilitate metastasis when upregulated, relating it to TNBC patient survival.

OTR expression differences between breast tumour tissue and normal breast tissue have been inadequately studied due to the problematic protein-based detection methods (lack of well-validated selective antibodies) [21,49]. Only a single study so far indicated that OTR expression is lower in tumour tissue (>11 fold at mRNA level and >2 fold at the protein level, n = 4) than in normal contralateral breast tissue [39]. According to the Pan-Cancer Analysis of Whole Genomes project [50], tissue adjacent to breast adenocarcinoma (distinct from healthy normal and tumour tissues) appears to have the highest OTR levels (adjacent > normal > adenocarcinoma). This study took advantage of the increasing gene expression data generated by a variety of high-throughput hybridisation array- and sequencing-based techniques (e.g., RNA-seq and ChIP-seq), accessible through databases such as the GEO database [23]. The OTR gene expression analysis of the breast tissue of several published datasets (Table 1) confirmed a higher OTR expression in tumour-adjacent tissue than in tumour tissue (Figure 1B,C).

There exist several potential reasons why OTR overexpression is observed in adjacent tissue but not in tumour tissue. First, in the female breast, OTRs are typically located on the cell membrane cells of the basal cell layer, which mainly consists of myoepithelial cells and undifferentiated cells, with very few epithelial cells being OTR-positive. However, most breast tumours arise from the epithelial cells [51], which could be a main reason for the high OTR levels observed in adjacent tissue. Another explanation could be that tumour-adjacent tissue is enriched for stroma pathways [52] and undergoes EMT [53], as increased OTR levels are also observed in the stroma in EMT and in fibroblast-to-myofibroblast transdifferentiation during the development of adenomyosis [54]. Other cancer development theories might also help to explain this adjacent-specific OTR overexpression. For example, it could relate to the tumour microenvironment in which tumour-secreted factors influence surrounding tissue to promote tumour invasion or metastasis [52]. OTR, unlike other members of the GPCR superfamily, can undergo dramatic and cell-specific up- and down-regulation [55]. For instance, OTR is upregulated in the uterine smooth muscle cells during gestation and mammary gland myoepithelial cells during lactation, resulting in higher OT sensitivity [56]; serum deprivation (a similar situation to nutrition deprivation in the solid tumour) of Hs578T breast cancer cells results in loss of OTR expression and OT-induced intracellular calcium accumulation, while serum restoration recovers OTR expression and responsiveness to OT [19]. Something similar might occur during breast tumour growth, with a changing tumour microenvironment affecting OTR expression, thereby modulating tumour growth, migration and/or invasion.

Based on the OTR-expression analysis, it was hypothesised that OTR expression could be linked to changes in the tumour microenvironment in preparation for metastasis. This hypothesis is supported by a mouse study using MDA-MB-231 cells, where the OTR levels were higher in metastasised tumour cells than in the corresponding primary xenograft tumour cells (Figure 2C) [27], as well as by this study’s transwell migration assays, where selected and amplified migrated T2 and T4 cells had ~2.1-fold and ~2.5-fold higher OTR expression than the parental control cells cultured in normal medium with 10% FBS (Figure 2B). These results align well with recent findings in a transgenic mouse model of OTR overexpression where OTR overexpression in the tumour microenvironment promotes mammary-specific tumour growth and metastasis [57]. Patients with metastatic TNBC usually have short progression-free survival (median 3–4 months) after the failure of first-line chemotherapy [58]. The link of high OTR expression to a higher likelihood of metastasis (which is associated with worse prognosis [59]) also aligns well with the patient survival data, where high OTR expression correlates with reduced survival of patients with TNBC (Figure 2F).

EGF is one of the most abundant growth factors, promoting EMT during metastasis by binding to EGFR on the plasma membrane of cancer cells. EGF enhanced cell migration of OTR-overexpressing MDA-MB-231 cells compared to control cells (Figure 4A), suggesting that the increased EGF sensitivity is linked to high OTR levels. OTR overexpression might affect EGFR expression, and crosstalk between EGFR and GPCRs has become increasingly well documented in different cellular systems, representing a prevalent mechanism of how GPCRs can regulate cell growth, migration, and invasion [60].

OT/OTR and EGF/EGFR signalling both play critical roles during mammary gland development, and OTR signalling often overlaps with EGFR pathways, supporting the possibility of interactions of OTR and EGFR signalling in breast cancer too. For example, OTR can induce either transient activation (<30 min) and translocation of EGFR/ERK1/2, thereby stimulating cell proliferation, or lead to a sustained EGFR/ERK1/2 activation (>3 h), thereby inhibiting proliferation in HEK293 and MDCK cells [61,62]. OTR activation can also promote long-term potentiation in hippocampal CA1 synapses by enhancing EGFR-mediated local translation of protein kinase Mζ [63] and inducing COX2 expression and PGF2 production in bovine endometrial epithelial cells involving EGFR transactivation [64], supporting the OTR–EGFR link. OTR and EGFR also share common signalling pathways, including the ERK1/2 and mTOR pathways that promote cell migration [12,60]. This study describes the first evidence of OTR expression affecting EGF-induced cancer cell migration capabilities and signalling. In OTR-overexpressing cells, ERK1/2-RSK-rpS6 signalling and cell migration still occurred, even in the presence of the EGFR-specific inhibitor gefitinib. Mechanistically, this appears to occur through EGF-mediated signalling via both EGFR and OTR to facilitate cell migration, since EGF activated not only EGFR but also induced intracellular signalling transduction and cell migration via an EGFR-independent but OTR-dependent mechanism (Figure 4 and Figure 5). Such EGFR-bypass signalling mechanisms have been linked to drug resistance in cancer cells in the past [65,66], and increased levels of OTR might also contribute to such resistance development for EGFR-targeted treatment in breast cancer patients.

The further downstream MEK1/2 inhibitor U0126 completely blocked ERK1/2-RSK-rpS6 phosphorylation and efficiently reduced cell migration (Figure 5C,D), supporting an EGF- and OTR-overexpression-associated signalling pathway via ERK1/2 activation that is independent of EGFR. Considering that rpS6 is a downstream effector of both the ERK and the mTOR pathways linked to cancer development and metastasis [12,41,42,46], the mTOR-selective inhibitor rapamycin was used to investigate the involvement of the mTOR pathway in the EGF-mediated signalling transduction. Inhibition of the mTOR pathway via rapamycin in the OTR-overexpressing MDA-MB-231 cells blocked rpS6 phosphorylation and reduced cell migration (Figure 5C,D), suggesting an involvement of the mTOR pathway that is independent of the ERK1/2 pathway, but to a minor degree (lower inhibition efficacy of rapamycin on cell migration than U0126). Taken together, these results support a dominant involvement of the ERK1/2-RSK-rpS6 pathway in enhancing EGF-induced cell migration in MDA-MB-231 cells.

Interestingly, this OTR-dependent pathway was not linked to classical ligand-receptor activation via G_q_ or G_s_ protein, as demonstrated by the absence of any signalling effects when the MDA-MB-231 cells were treated with OT or other OTR-specific ligands (Figure 3C,D), indicating compromised G_q_ and G_s_ functionality. The involvement of the V_1a_R/V_1b_R-mediated G_q_ pathway and the V_2_R-mediated G_s_ pathway was excluded since OT would activate these receptors at the tested concentrations (Figure 3C,D) [35]. MDA-MB-231 cells also have no OT mRNA expression [67]; exogenous OT did not affect cell migration (Figure 4C) or the metastasis markers (Figure 4D,E); nor did OT induce ERK1/2, RSK, and rpS6 activation (Appendix A). Such a compromised OT/OTR signalling system could however be of advantage to cancer cells considering the reported protective effects of OT/OTR signalling [39,68,69]. Based on all these observations, the proposed signalling transduction model is presented in Figure 6.

Together, these findings demonstrate that high OTR expression is linked to increased MDA-MB-231 cell migration and reduced survival of TNBC patients; however, how EGF triggers the intracellular signal transduction in an OTR-dependent manner remains an open question and further research is needed to provide more detailed mechanistic insights into the interactions and signalling pathways of the OTR and EGFR pair, including whether and how OTR and EGFR could regulate the expression of each other. This study furthermore raises a critical point concerning whether OTR’s function is compromised also in other breast cancer cell lines, something that should be investigated early on when studying OTR in breast cancer. Of note, only the main G_q_ and G_s_ pathways were investigated; the G_i_ and β-arrestin pathways were not examined.

## 5. Conclusions

In summary, this study provides first evidence that high OTR levels can promote metastasis in TNBC, correlating with reduced patient survival. The underlying mechanisms are linked to enhanced EGF sensitivity and the activation of the ERK1/2-RSK-rpS6 pathway promoting cell migration. The results support the existence of an EGF-mediated downstream ERK1/2-RSK-rpS6 signalling pathway that bypasses EGFR in the presence of high OTR expression and compromised OTR signalling, which could be a cancer growth or survival adaptation. These are interesting new observations that should be verified across other TNBC cell lines as well as in vivo. These findings could have several implications towards the improved clinical management of TNBC, including using high OTR expression as a prognostic biomarker or as a target for therapeutic interventions to prevent or reduce TNBC metastasis.

## Figures and Tables

**Figure 1 biomedicines-10-01595-f001:**
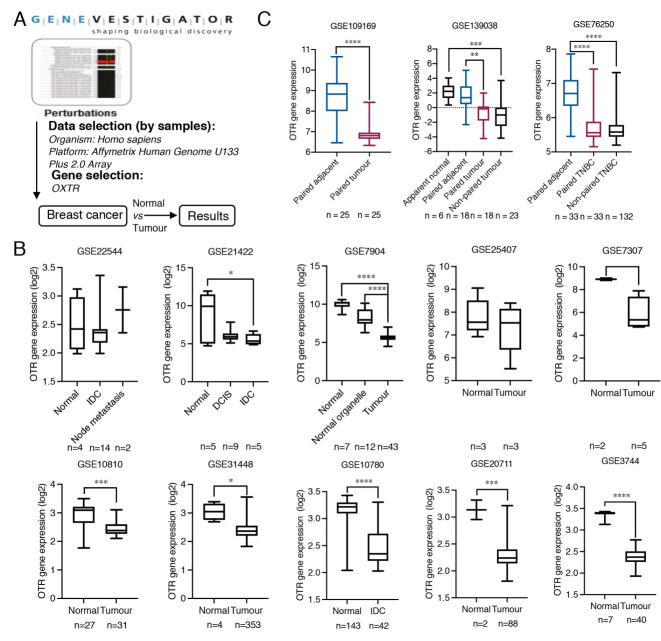
Oxytocin receptor gene expression analysis in normal, adjacent and, tumour breast tissue. (**A**) The workflow of the strategy to find OTR expression alteration specific to breast cancer using the Perturbations tool from Genevestigator. (**B**) Detailed OTR gene (*OXTR*) expression data (log2 transformed) comparing tumour and normal breast tissues from breast cancer RNA-Seq expression datasets, based on Affymetrix Human Genome U133 Plus 2.0 Array platform, extracted from Genevestigator. DCIS, ductal carcinomas in situ, IDC, invasive ductal carcinomas. (**C**) OTR gene expression analysis of paired breast tumour and adjacent breast tissue. The datasets are based on various sequencing platforms and presented as expression data obtained from the GEO database without log2 transformation. TNBC, triple-negative breast cancer. All human patient data are expressed as a min to max boxplot. The Mann–Whitney test was used to analyse non-paired samples from two groups. One-way ANOVA analysis followed by Tukey’s multiple comparison test was used to compare datasets containing more than two groups. A paired *t*-test was used to analyse paired tumour and normal or adjacent samples. *, *p* < 0.05, **, *p* < 0.01, ***, *p* < 0.001, ****, *p* < 0.0001.

**Figure 2 biomedicines-10-01595-f002:**
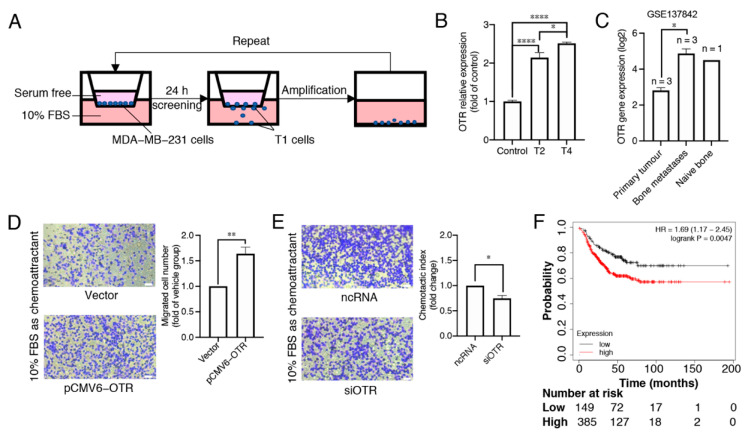
High oxytocin receptor expression correlates with increased metastasis potential in triple-negative MDA-MB-231 breast cancer cells. (**A**) The workflow of the transwell migration assay showing the separation of migrated cells (T1 cells). After amplification under normal cell culture conditions, the amplified T1 cells were admitted to the next cycle of migration and amplification. This cycle was repeated four times, and the T2 and T4 cells were collected for OTR RNA expression analysis. (**B**) qPCR results for OTR RNA expression in different subtypes of MDA-MB-231 cells, the housekeeping gene glyceraldehyde-3-phosphate dehydrogenase (GAPDH) was used as internal control. The data are presented as fold changes compared to the control cells (wild-type MDA-MB-231 cells). Data are means ± SD of one experiment performed in triplicate. (**C**) OTR RNA expression (log2 transformed) of different groups from the microarray dataset GSE137842, which was derived from the published in vivo study mimicking human bone metastasis in mice using MDA-MB-231 cells [27]. (**D**,**E**) Representative images (10× objectives, scale bar (bottom right, white) = 41.86 μm) and migrated cell number (fold changes of control group) of MDA-MB-231 cells with OTR upregulation/downregulation in the transwell migration assay obtained from experiments conducted at the same time (n = 3 independent experiments with triplicates). Vector group, the control MDA-MB-231 cells transfected with the vector plasmid; pCMV6-OTR, MDA-MB-231 cells transfected with the pCMV6-OTR plasmid; ncRNA, the control MDA-MB-231 cells transfected with the non-targeting control RNA; siOTR, MDA-MB-231 cells transfected with OTR-targeting siRNA SMARTpool. (**F**) Relapse-free survival (RFS) analysis of OTR in TNBC patients (restrict patient population to ‘triple-negative’: IHC ER^-^, IHC PR^-^, array HER2^-^) from KM-plotter. The patients were divided into high- and low-expression groups using the best cutoff. ‘Only JetSet best prob set’ was used as the probe set option, and ‘exclude biased arrays’ was selected for array quality control purposes. Statistical analysis for (**B**) included ANOVA analysis followed by Tukey’s multiple comparisons test. Statistical analysis for (**C**) included a paired *t*-test. Statistical analysis for (**D**,**E**) included a Student’s *t*-test. *, *p* < 0.05; **, *p* < 0.01; ****, *p* < 0.0001.

**Figure 3 biomedicines-10-01595-f003:**
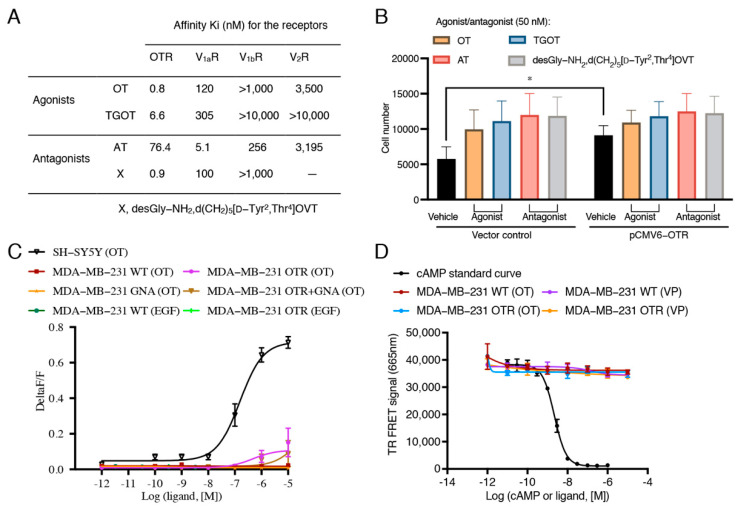
Effects of oxytocin receptor agonists and antagonists on migration ability and effects of oxytocin on intracellular calcium or cAMP accumulation in MDA-MB-231 cells. (**A**) Affinity information of OTR agonists and antagonists selected for the cell migration study obtained from references [28,29]. (**B**) Effects of OTR agonists and antagonists (50 nM) on cell migration of MDA-MB-231 cells with normal OTR expression (vector group) and with OTR overexpression (pCMV6-OTR group). 10% FBS was used as the chemoattractant. The same volume of H_2_O was used as vehicle control of peptide treatments. The concentrations of the ligands were set to 50 nM to ensure OTR selectivity for the OTR-selective ligands. Statistical analysis for (**B**) included ANOVA analysis followed by Tukey’s multiple comparisons test (n = 3 independent experiments with triplicates). *, *p* < 0.05. OT, oxytocin; AT, atosiban; OTR, oxytocin receptor; V_1a_R, vasopressin receptor 1a; V_1b_R, vasopressin receptor 1b; V_2_R, vasopressin receptor 2. (**C**) Representative OT concentration–response curve measuring calcium signals in SH-SY5Y cell line, wild-type (WT), OTR, and/or GNA transfected MDA-MB-231 cells treated by OT (1 pM–10 μM) and EGF (0.02 ng/mL–20 ng/mL). SH-SY5Y cell line with native OTR expression and good G_q_/calcium response was used as the positive signal control for the FLIPR assay. OTR and GNA15-overexpression was achieved by transient transfection of OTR- or GNA15-expressing plasmids or by co-transfection of OTR- and GNA15-expressing plasmids. (**D**) cAMP standard curve for the LANCE Ultra cAMP signalling assay kit and representative OT/VP concentration–response curve measuring cAMP accumulation for WT or OTR-overexpressing MDA-MB-231 cells. Vasopressin (VP) was included to control for any potential involvement of V_2_R-related cAMP signalling (VP more potent at V_2_R than OT). Data in **C** and **D** are means ± SD of one representative experiment (n = 3 in total) performed in triplicate.

**Figure 4 biomedicines-10-01595-f004:**
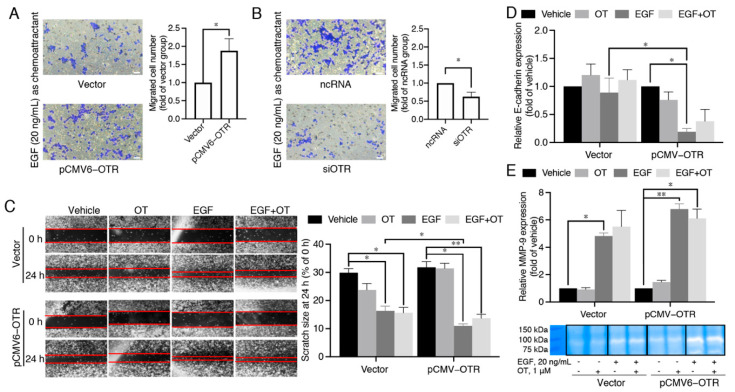
High oxytocin receptor expression correlates with high sensitivity to EGF-stimulated migration. (**A**,**B**) Effects of OTR overexpression or knock-down on EGF-guided MDA-MB-231 cell chemotaxis obtained from experiments conducted at the same time (10× objectives, scale bar (bottom right, white) = 41.86 μm, n = 3 independent experiments with triplicates). Vector group, the control MDA-MB-231 cells transfected with the vector plasmid; pCMV6-OTR group, MDA-MB-231 cells transfected with the pCMV6-OTR plasmid; ncRNA group, the control MDA-MB-231 cells transfected with the non-targeting control RNA; siOTR, MDA-MB-231 cells transfected with OTR-targeting siRNA SMARTpool. (**C**) Effects of OTR overexpression or knock-down on cell migration tested by a scratch wound healing assay. Photos were taken using a Nikon Ti-U inverted fluorescence microscope at 0 h and 24 h after scratches were made (20× objectives). (**D**) mRNA expression of the metastasis marker E-cadherin in MDA-MB-231 cells with or without OTR overexpression treated by EGF (20 ng/mL) and/or OT (1 μM). (**E**) Comparison of MMP-9 activity in MDA-MB-231 cells with or without OTR overexpression treated by EGF (20 ng/mL) and/or OT (1 μM) determined by a gelatin zymography assay. The bands were derived from the same gel. The bar graph quantifies MMP-9 bands by densitometric analysis. Vehicle group, MDA-MB-231 cells without EGF/OT treatment. Statistical analysis for (**A**,**B**) included the Student’s *t*-test. Statistical analysis for (**C**–**E**) included ANOVA analysis followed by Tukey’s multiple comparisons test. n ≥ 3 independent experiments, *, *p* < 0.05, **, *p* < 0.01.

**Figure 5 biomedicines-10-01595-f005:**
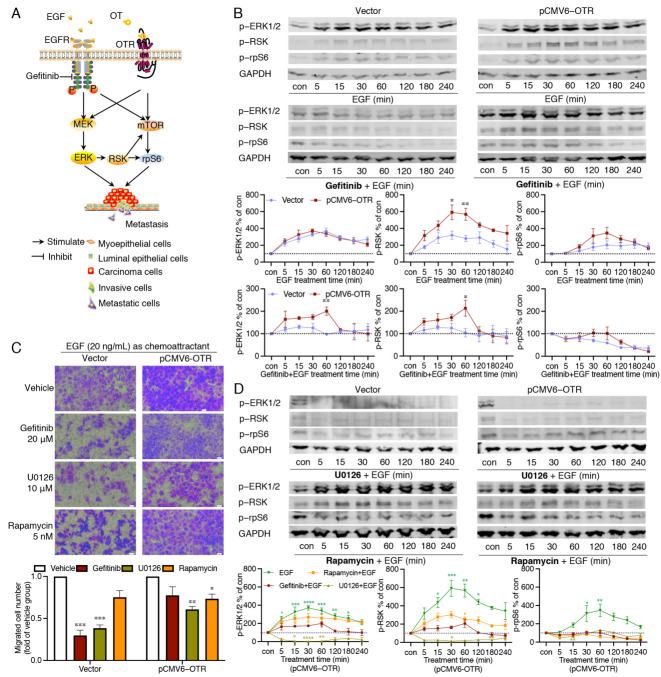
Effects of EGF on ERK1/2, RSK and rpS6 phosphorylation and migration ability in OTR-overexpressing cells. (**A**) EGFR and OTR independently induce ERK1/2 and mTOR signalling linked to cancer metastasis. (**B**) Representative Western blot images and quantification of phosphorylated ERK1/2, RSK, and rpS6 relative to GAPDH in EGF-treated cells with or without gefitinib (20 μM) pre-treatment, the cells included control (vector) and OTR-overexpressing (pCMV6-OTR) MDA-MB-231 groups. (**C**) Representative images (20× objectives, scale bar (bottom right, white) = 21.18 μm) and quantification of migrated MDA-MB-231cells (with or without OTR overexpression) attracted by EGF (20 ng/mL) and treated with inhibitors (n = 2 independent experiments with triplicates). (**D**) Representative Western blot images of phosphorylated ERK1/2, RSK, and rpS6 in control (vector group) or OTR-overexpressing (pCMV6-OTR group) MDA-MB-231 cells pre-treated with U0126 (10 μM) or rapamycin (5 nM), with quantification of the phosphorylated ERK1/2, RSK, and rpS6 in control MDA-MB-231 cells. Con, control samples from cells without treatment, which reflected the base line of the protein activation status in (**B** and **D**). The quantitation of ERK1/2, RSK, and rpS6 phosphorylation were normalised relative to the level of GAPDH (n = 3 independent experiments). Vector group, MDA-MB-231 cells transfected with the vector plasmid, which were used as control cells without OTR overexpression; pCMV6-OTR group, MDA-MB-231 cells transfected with the pCMV6-OTR plasmid for OTR overexpression. Data were analysed using ANOVA analysis followed by Tukey’s multiple comparisons test. *, *p* < 0.05, **, *p* < 0.01, ***, *p* < 0.001, ****, *p* < 0.0001, vs. vehicle group.

**Figure 6 biomedicines-10-01595-f006:**
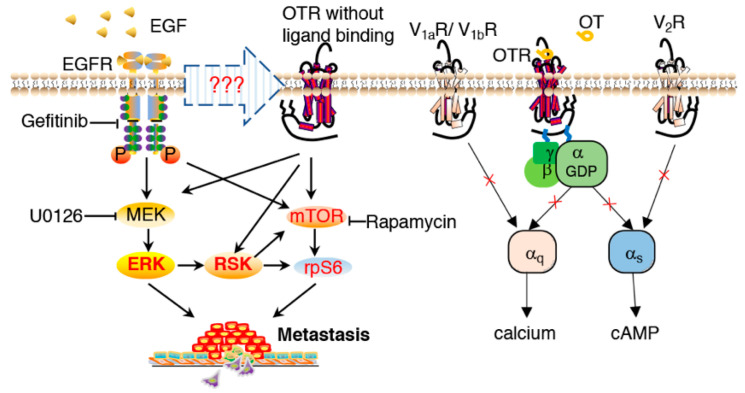
EGF-mediated signal transduction involving the oxytocin receptor in promoting metastasis. Normal OT/OTR signalling via G_q_ or G_s_ was compromised in MDA-MB-231 cells. Involvement of the V_1a_R/V_1b_R-mediated G_q_ pathway and V_2_R-mediated G_s_ pathway were excluded. In OTR-overexpressing MDA-MB-231 cells, EGF induced ERK1/2-RSK activation via both EGFR and OTR. The mTOR pathway was also involved in EGF-stimulated rpS6 activation, independent of ERK1/2 activation.

**Table 1 biomedicines-10-01595-t001:** Overview of the datasets containing both breast cancer and normal breast tissues identified in Genevestigator and in the GEO database.

Datasets Containing Breast Cancer and Normal Breast Tissues Identified Using Genevestigator in Figure 1B			
GEO Accession	Overall Design and Sample Information(Affymetrix Human Genome U133 Plus 2.0 Array Platform, OTR Gene ID, 206825_at)		
GSE22544	16 invasive ductal carcinomas samples (including 2 node metastasis samples) analysed with the U133 Plus 2.0 array compared to 4 normal control samples		
GSE21422	Dataset including 5 healthy tissue samples, 9 ductal carcinomas in situ, and 5 invasive ductal carcinomas		
GSE7904	62 samples including 43 tumours, 7 normal breast, and 12 normal organelles		
GSE25407	3 examples of Stage-I breast tumour and 3 samples of breast reduction mammoplasty tissue were expanded as explant cultures for RNA extraction and hybridisation to Affymetrix microarrays		
GSE7307	Affymetrix human U133 plus 2.0 array was used to transcriptionally profile both normal and diseased human tissues representing over 90 distinct tissue types (herein, breast tissues analysed only)		
GSE10810	58 samples including 31 tumours and 27 controls; some of the samples are paired		
GSE31448	Tumour tissues from 353 patients with invasive adenocarcinoma who underwent initial surgery. Dataset also includes 4 normal breast samples		
GSE10780	143 histologically normal breast tissues and 42 invasive ductal carcinoma tissues		
GSE20711	Dataset includes 2 normal breast tissues and 88 breast tumour tissues		
GSE3744	Dataset includes 7 normal breast tissues and 40 breast tumour tissues		
**Datasets from the GEO database containing paired adjacent and breast tumour tissues in Figure 1C ^#^**			
**GEO accession**	**Samples**	**Platform**	**OTR gene ID ^a^**
GSE109169	25 sets of paired adjacent/tumour breast tissue specimens	GPL5175 [HuEx-1_0-st] Affymetrix Human Exon 1.0 ST Array [transcript (gene) version]	2661992
GSE139038	65 samples including 41 breast tumours, 18 adjacent tissues [paired normal], and 6 apparently normal tissues from breasts operated on for non-malignant conditions	GPL27630 Print_1437 (Block_Column_Row IDs)	7_14_20
GSE76250	198 samples including 165 TNBC tissues and 33 paired adjacent breast tissues	GPL17586 [HTA-2_0] Affymetrix Human Transcriptome Array 2.0 [transcript (gene) version]	TC03001145.hg.1

^a^, these studies were retrieved from the GEO database, and the OTR ID was different in different sequencing platforms. ^#^, tumour tissue was defined as having over 70% (GSE139038) or 80% (GSE76250) tumour cells, paired normal and apparently normal had to be morphologically normal with no tumour cells (GSE139038).

## Data Availability

Data are contained within the article or supplementary material.

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
