# Peer review of "High Oxytocin Receptor Expression Linked to Increased Cell Migration and Reduced Survival in Patients with Triple-Negative Breast Cancer"

_biomedicines, 2022, doi:10.3390/biomedicines10071595_

Round 1
Reviewer 1 Report
Liu et al. biomedicines-1744433" High oxytocin receptor expression linked to increased cell migration and reduced survival in patients with triple-negative breast cancer" is a valuable review paper that shows EGF induces ERK1/2-RSK activation via both EGFR and OTR in OTR-overexpressing MDA-MB-231 cells. Also, the authors show that the mTOR pathway is involved in EGF-stimulated rpS6 activation, independent of ERK1/2 activation. However, one point was difficult for the reviewer to understand. The reviewer hopes that providing more information (described below) will help improve this study's quality.
As shown in Figure 5, the Authors indicate that Phosphorylated ERK1/2 (p-ERK1/2), RSK 403 (p-RSK), and rpS6 (p-rpS6) increased in OTR-overexpressing MDA-405 MB-231 by treatment of EGF. In this part, the reviewer thought about the possibility of increasing EGFR in OTR-overexpressing MDA-405 MB-231. Does the expression of EGFR not change between OTR-overexpressing and control cells? The reviewer thinks that the authors should indicate this point.
Minor comment
The panels in Figure 5 are too small to read the text in the panels. Therefore, the reviewer thinks that figures should be resized to make them easier to read.
Reviewer 2 Report
The paper is well written and the results are extensively commented. The authors gave new insights on the role of oxytocin as promoter of metastasis in TNBC, increasing the clinical relevance for personalized treatment.
Reviewer 3 Report
The current manuscript explores the expression of oxytocin receptor (OTR) on the migration of triple-negative breast cancer (TNBC). OTR was selected by Genevestigator, and further evaluated by MDA-MB-231 cell line, the authors observe that OTR was associated with cancer metastasis through EGF-mediated ERK1/2-RSK-rpS6 pathway.
Major comments:
The introduction can be slightly expanded, and transitions between sections smoother. (1)Discuss 5-year relative survival rate of female breast cancer patients (2) Provide more information on the signaling pathway of migration in cancers.
M & M: (1) To illustrate how to grouping and evaluate the differentially expressed gene by Genevestigator. (2) Simplify common experimental methods.
Results: (1) To reduce artificial bios and increase data reliability, it is recommended that experiment Fig 2D and 2E should be done at the same time. (2) it is recommend to evaluate EGFR and the signaling in “T2” and “T4” cell line.
Discussion: (1) Please further discuss the signaling pathway between EGF and OTR. More and more evidences reveal that the 2 receptors are associated in breast cancer. Therefore, please insert in Bibliography more recent references. (2) Please discuss any study limitations.
Round 2
Reviewer 1 Report
The second revised paper seems to be improved well and should be worth publishing in this journal.
Reviewer 3 Report
The manuscript is ready for publication after revision.